



# The selection of directional sectors for the analysis of extreme wind speed

Pedro Folgueras[1], Sebastián Solari[2], and Miguel Ángel Losada[1]

[1]Group of Environmental Fluid Dynamics (IISTA). University of Granada. Avda. del Mediterráneo, s/n, 18006, Granada (Spain)
[2]Instituto de Mecánica de los Fluidos e Ingeniería Ambiental. Universidad de la República. Julio Herrera y Reissig 565, 11300 Montevideo (Uruguay)

**Correspondence:** Pedro Folgueras (folgueras@ugr.es)

**Abstract.** This paper presents a rational method for the selection of the most suitable directional sectors in the analysis of extreme wind loads on structures. It takes into consideration the main sources of uncertainty stemming from sector selection, and leads to the definition of independent and statistically homogeneous directional sectors.

This method is applied to the selection of directional sectors for the calculation of the design wind speed of a structure
located at the mouth of the Río de la Plata. The results in the estimated reliability and costs were compared to those obtained with conventional engineering methods revealing significant differences. It was found that the proposed method is a simple and objective tool for the selection of directional sectors, which comply with the working hypothesis of the directional models and offers better guarantees for dimensioning than the use of more traditional engineering approaches for sectorial division.

## 1 Introduction

Wind directionality effects have a well recognized impact on the characteristics of the extreme wind loads of structures. The methods for dealing with it usually involves the division of available data into sectors (whose statistical behavior is assumed to be homogeneous) and the evaluation of the extreme behavior of the wind velocity in each of them. The implicit decisions involved in this procedure (and its uncertainty) include: (i) the identification of extreme values; (ii) the selection of the optimal model for data fitting; (iii) the definition of the directional sectors for calculation; and (iv) the characterization of the dependence
between directional extremes. From all of them, the selection of the sectors, which is the subject of this article, has received the least attention.

Wind tunnel laboratories and building codes have developed multiple methods in order to consider the influence of directionality on the estimation of extreme wind speeds and wind-induced quantities, such as, the "up-crossing method", the "worst case" method, the "storm passage method", etc. (see, for example., Irwin et al. 2005; Isyumov et al. 2014). Among them, the
"sector-by-sector" approaches attempt to produce directional wind speeds, or directional wind-speed multipliers, for a discrete number of defined wind directions. The model of extreme values is fit in each sector separately assuming data allocated in sectors is independent. When the directional wind speeds are combined with the measured structural response coefficients, the largest resulting response from any direction is deemed to be an appropriate design value (Holmes, 2015).





Recently, Zhang and Chen (2015, 2016) propose a methodology to estimate the probability distribution of the load responses of structures under extreme wind conditions, which is an extension of the probabilistic methods of Cook and Mayne (1979, 1980). This method allows the study of both, the directional extreme winds and the directional distribution of the response coefficients, separately. If the directional response coefficients are poorly correlated and if they can be estimated from wind tunnel tests for a particular structure, then the main challenge in applying the method is the calculation of the multivariate distribution of the extreme wind directional velocity.

Using a priori defined divisions to this end results, in general, in correlated directional sectors. This complicates the use of these methods, since the dependence structure between the extreme directional values of wind speeds must be modeled using any of the existing approaches (e.g., Simiu et al. 1985; Coles and Walshaw 1994; Solari and Losada 2016). It also poses other potential issues such as the lack of enough data in some sectors or the existence of non-homogeneous populations, among others. Nevertheless, engineering methods generally opt for the use of simple criteria, mainly based on the definition of sectors of fixed amplitude, oriented according to the cardinal directions (see, e.g.Mayne 1979; Cook 1982, 1983; Cook and Miller 1999, which use 30 deg sectors). Also, current regulations and guidelines (API, 2000; DNV, 2010; ISO, 2005) for the design of offshore structures (which consider wind directionality but also other directional variables, such as waves and currents), deal with the division of the compass into sectors. The API Recommendations suggest taking the main direction of the agent as the reference direction whereas the DNV leaves this decision to the engineer. As an alternative to these approaches, ISO (2005) proposes the use of naturally defined sectors based on the directionality inherent in the data measured or obtained in reanalysis. However, this guideline does not provide any specific criteria that can be used to implement this approach.

In this work an alternative methodology that defines the distribution of the extreme wind directional velocity in a non-arbitrary manner is proposed. Both, intrasectorial homogeneity and intersectorial independence conditions are imposed, among others, to obtain the directional sectors. Unlike previous approaches, this methodology results in uncorrelated sectors, so it is not necessary to use dependence models (e. g. copulas) and allows to approximate the multivariate (directional) extreme distribution simply as the product of the marginals. In addition, this method assures directional sectors that contain data in consonance with the working hypotheses of the directional model for the extremes.

This methodology was applied to the study of extreme values of wind speed at the study zone of the mouth of the Río de la Plata, where directional effects are of particular importance. The effect of directional sector selection on the design wind speed of a structure was also estimated. These calculations and their consequences for project design reliability were compared with the results obtained with traditional engineering methods based on the use of divisions with equal size sectors and a northern direction of origin.

The rest of this paper is organized as follows. After defining the research problem, Section 2 delimits the framework and specifies the main sources of uncertainty considered. The methodology for the selection of directional sectors is then explained. The case study in Section 3, describes the wind characteristics in the study zone, followed by the quantification of the impact of sector selection, based on indicators for each source of uncertainty. These results are compared to those obtained with conventional engineering methods. In Section 4, a simple example is used for illustrating the potential consequences of the



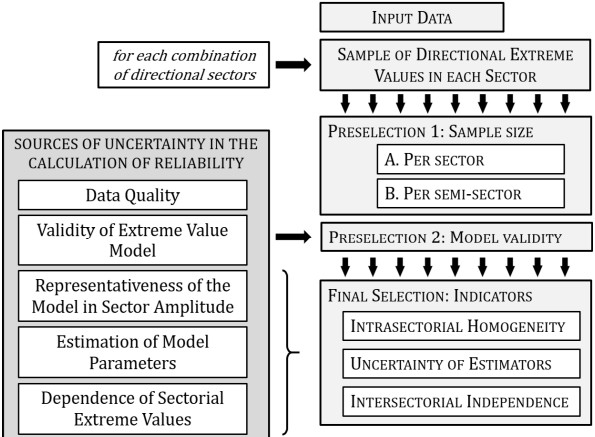

**Figure 1.** Methodology for sector definition, based on the sources of uncertainty in the calculation of reliability in a system subjected to directional extreme values.

selection of directional sectors for project design reliability. Finally, Section 5 presents the main conclusions that can be derived from this research.

## 2   Methodology for the specification of directional sectors

### 2.1   Problem statement

The selection of calculation sectors affects the estimates of directional extreme values, which may impact on the evaluation of project costs and structural reliability. The main factors that influence the result are the following: (1) the procedure followed to identify the extreme events of the sector samples; (2) the validity of the model used to characterize extreme behavior; (3) the goodness of parameter estimation; (4) the capacity of each model to represent extreme behavior in the total amplitude of the corresponding sector; (5) the validity of the dependence model between extreme values in different sectors. All of these factors

are in turn conditioned by the quantity of available data and their directional distribution.

For the selection of calculation sectors, this paper describes a procedure that considers the main sources of uncertainty stemming from the choice of sectors. Firstly, the candidate divisions are limited to those whose sectors are compatible with the selected model of extreme values, and which have a certain minimum quantity of information. Secondly, the consequences of this selection are evaluated for each division by means of indicators that characterize the intrasectorial homogeneity of

the samples, the uncertainty of the estimates of directional extreme values, and their intersectorial independence. Finally, the division with the best overall behavior is selected, based on the set of indicators. This methodology is outlined in the flowchart in Figure 1.



### 2.1.1 Requirements for the preselection of candidate division

Extreme events are isolated in each sector for different possible angular divisions by means of any appropriate technique, such as peaks over threshold, block maxima (Coles, 2001), weather patterns (Solari and Alonso, 2017), etc. Divisions containing sectors that do not meet the following two requirements are excluded: (1) the division must have the quantity of data minimally

necessary to test the validity of the models used for describing the extreme behaviour; (2) the data in each division must also be compatible with these models.

Regarding the first requirement, the minimum acceptable quantity of data in each sector (or semi-sector) should be such that the probability of a Type II error (a false negative finding) in the statistical hypothesis tests that are used in the proposed method is less than a given value $\beta$. For this purpose, the power curves of these tests, which relate $\beta$ to the minimum amount of data, are

used. The significance level $\alpha$ and an the effect size of these curves should be defined in consonance to the problem under study. Regarding the second requirement, the extreme events should not be incompatible with the selected model of extreme values. Compliance with this requirement is evaluated by means of bilateral hypothesis testing, e.g., Anderson-Darling (Anderson and Darling, 1952) or Kolmogoroff-Smirnoff (Kolmogoroff, 1941; Smirnoff, 1939), with significance level $\alpha$.

### 2.1.2 Selection of the calculation sectors

The next step involves the evaluation of the consequences of sector selection on: (i) the intrasectorial homogeneity of the samples; (ii) the uncertainty of the estimates of directional values; and (iii) their intersectorial independence. To this end, the use of three indicators based on standard statistical analysis and which are measured on a 0-1 scale, is proposed. This approach is also compatible with the use of other indicators specific to the problem under consideration.

The first indicator characterizes the variability of the statistical behavior of extreme events along the arc of each sector.

Significant discrepancies between the subsamples of a sector can indicate the presence of different populations, which is incompatible with the hypotheses of the model used. The second indicator reflects the uncertainty of the fit of extreme values by analyzing their asymptotic distribution. Finally, the third indicator evaluates the incompatibility of the sectors with the independence between sectorial extreme values. This independence occurs when each storm event is restricted to one sector and does not move to neighboring ones.

The division selected is the one that shows the best overall performance as reflected in the set of characteristics evaluated by the indicators. This leads to the creation of a new global indicator, which is a function of the other three, and which allows for sorting the candidate divisions according to the selected criterion.

## 2.2 Specification of indicators

### 2.2.1 Indicator of intrasectorial homogeneity

In order to compare the statistical homogeneity of extreme values in different regions of the same sector, the sector is divided into two subsectors of equal amplitude ($S_1$ and $S_2$). A generalized Kolmogoroff-Smirnoff (Smirnoff, 1939; Kolmogoroff,



1941) test is performed, which evaluates the degree of incompatibility of the two subsamples of POT events in virtue of the null hypothesis that both belong to the same population.

As an indicator of this characteristic in a sector, the p-value of the contrast is used. This indicates the probability, given the null hypothesis $H_0$ is true, that the KS test statistic, $D$, has a value that is greater than or equal to that given by the data. The

smaller the p-value, the grater the statistical incompatibility of the data with the null hypothesis, if the underlaying assumption used to calculate the p-values holds (Wasserstein and Lazar, 2016). The behavior of the sectors is evaluated by means of the geometric mean given by Eq. 1, where $d_m$ is the test statistic in each sector.

$$\overline{p_1} = \left\{ \prod_{m=1}^{M} Prob\left[ D \geq d_m | H_0 \right] \right\}^{1/M} \tag{1}$$

### 2.2.2 Indicator of uncertainty of the estimators

This indicator gives a measure of the uncertainty stemming from the choice of sectors, in the estimations of extreme values. This uncertainty is characterized by analyzing the asymptotic distribution of the extreme values within the context of Delta method hypotheses (e.g. Coles 2001). For this purpose, in each sector, the probability corresponding to the intervals defined by means of the estimated extreme value and a discrepancy $\pm\varepsilon_0$ is evaluated. The performance of the set of sectors is calculated by means of the geometric mean of the results obtained for each one, as shown in Eq. 2.

$$\overline{p_2} = \left\{ \prod_{m=1}^{M} \left[ 1 - 2\Phi_{0,1}\left( -\frac{\varepsilon_0}{\sigma_{E_m}} \right) \right] \right\}^{1/M} \tag{2}$$

Where $\Phi_{0,1}(\cdot)$ is the value of the standard normal distribution function, $\sigma_E$ is the standard deviation of the estimator and the discrepancy $\varepsilon_0$ (%) is a parameter that must be previously defined.

### 2.2.3 Indicator of intersectorial independence

In the case of completely independent sectors, the following relation is verified:

$$Pr\left[ Y \leq x \right] = Pr\left[ X_1 \leq x \right] \cap Pr\left[ X_2 \leq x \right] \cap \cdots \cap Pr\left[ X_M \leq x \right] = \prod_{m=1}^{M} Pr\left[ X_m \leq x \right] \tag{3}$$

where $Y$ is the annual maximum value of omnidirectional variables, and $X_m$ is the annual maximum value of directional variables in sector $m$.

The *p-value* of the Kolmogoroff-Smirnoff test (Smirnoff, 1939; Kolmogoroff, 1941) is used as an indicator of the incompatibility of the omnidirectional data with a model based on the independence of sectorial extremes corresponding to a given

division (Eq. 3). The null hypothesis $H_0$ is thus assumed, according to which omnidirectional annual maximum values conform to this model. The sample of omnidirectional annual maximums is checked against the distribution obtained by multiplying



directional distributions that are fit to the data. The value of the indicator, $\overline{p_3}$, is given by Eq. 4, where $d$ is the test statistic, corresponding to the sample of omnidirectional annual maximums.

$$\overline{p_3} = Prob[D \geq d | H_0] \tag{4}$$

### 2.2.4 Global indicator

In order to consider the previous indicators as a whole, the Euclidean norm defined in Eq. 5 was used.

$$\|\overline{p_i}\| = \sqrt{\overline{p_1}^2 + \overline{p_2}^2 + \overline{p_3}^2} \tag{5}$$

where $\overline{p_1}$, $\overline{p_2}$ and $\overline{p_3}$ are the indicators of the intrasectorial maximum homogeneity, minimum uncertainty in estimations, and intersectorial maximum independence, respectively. All are measured on the same scale 0-1 , where 0 represents the worst qualities, and 1, the best qualities. Consequently, the value of $\|\overline{p_i}\|$ is delimited by 0 and $\sqrt{3}$, where 0 represents the worst

value of the indicator and $\sqrt{3}$, the best value.

### 2.3 Outline of the procedure

The procedure involves the following steps:

1. Identification of extreme events per sector

2. Definition of requirements and conditioning factors (section 2.1.1)

(a) Requirements regarding data quantity

        (b) Requirements for the validity of the models by sectors

3. Preselection of the sets of sectors that meet these requirements

4. Selection of the calculation sectors (section 2.1.2)

        (a) Evaluation of the indicators in each candidate division (section 2.2)

(b) Selection of the set of sectors with the best overall indicator value

## 3   Case study

### 3.1   Description of the agent at the site

The study site is located in front of the mouth of the Río de la Plata [36° S, 55° O] on the east coast of South America between Uruguay and Argentina (Figure 2). The estuary there is one of the largest in the world and is of great interest from both a social





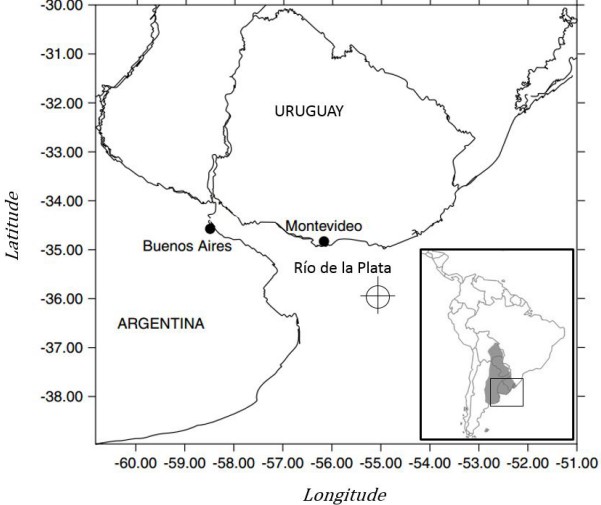

**Figure 2.** Location of the study site at the mouth of the Río de la Plata

and ecological perspective. Since it is also an extremely active zone of cyclogenesis, it has been the focus of much research (e.g., Framiñan et al. 1999; Guerrero et al. 1997; Solari and Losada 2016).

Atmospheric circulation in the area is controlled by the South Atlantic high-pressure system, which brings hot, humid air to the estuary. In addition, cold-air systems from this anticyclone bring masses of cold air to the zone approximately every four days. This means that wind direction frequently varies since northeasterly winds alternate with southeasterly winds every few days (Simionato et al., 2007).

Furthermore, intense storms, known as "sudestadas" [Southeast blows], often occur during the summer. These events are produced by anticyclonic cells from subtropical latitudes with strong southeasterly winds, loaded with humidity, which bring heavy rain to the estuary. The river's southeast alignment produces rough seas and meteorological tides. During the winter months, masses of cold air from the Antarctic anticyclone ("pamperos") blow from the southeast causing a considerable drop in temperatures.

## 3.2 Directional variability of extreme events

The research data used in this study come from reanalysis time series of the ERA-Interim program (Dee et al., 2011), belonging to the European Centre for Medium-Range Weather Forecasts (ECMWF). The variables extracted from the database are 10-minute average wind speed components U and V at a height of 10 meters and recorded at a rate of 6 hours. There were 37 years of data available from January 1979 until December 2015. The characteristics of these data (origin, sampling rate, duration and quality of the time series, etc.) add an initial uncertainty that should be considered for the estimation of extreme events. However, in order to focus the discussion on the proposed method, only the sources of uncertainty that arise from the process of directional discretization will be taken into account henceforth. rfrg





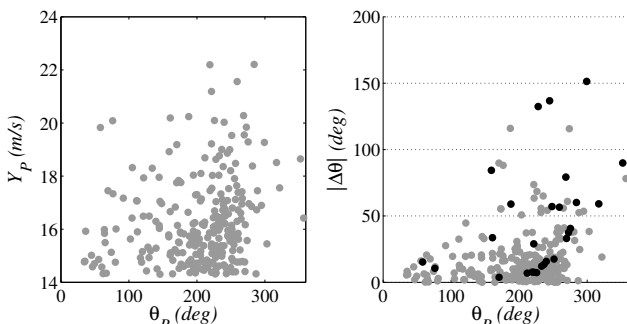

**Figure 3.** Left panel: Directional variability of the peak velocities of storm events. Right panel: Variability of the angular distance traveled by each storm.

Independent events were identified by applying the Peaks-Over-Threshold (POT) method with a time window of 5 days between storms, to the omnidirectional data. In this way, a total of 270 storms were isolated. Each storm was characterized by its maximum wind speed $Y_i^P$, the corresponding direction $\theta_i^P$ and its angular distance traveled $|\Delta\theta|$. The magnitude and frequency of occurrence of the variable (left panel of Figure 3) suggest that direction is a relevant covariable for the characterization of extreme events. The right panel of Figure 3 shows the maximum angular distance traveled by each storm depending on $\theta_i^P$. For this calculation, the time evolution curve of the direction of each storm event was reconstructed and the absolute maximum discrepancy between its values at any two moments, measured in the direction of movement, was evaluated.

There were significant variations in direction with regard to the values where the peaks occurred. More specifically, there were displacements greater than $45°$ in $12\%$ of the storm events, and one third of them (black dots) have maximum associated wind speeds higher than those in the $90^{th}$ percentile. This indicated that storm events can extend over more than one directional sector and, therefore, a potential dependence between the extreme values of neighboring sectors.

To take into account this directional dissipation in the extreme value modeling (Jonathan and Ewans, 2007), the sectorial samples of extreme events in terms of the peak value of each storm in the sector under consideration were characterized. The initial definition of POT events was based on the same threshold obtained from omnidirectional data, which was selected with the method in Solari et al. (2016).

### 3.3 Analytical framework

The GEV model is often chosen to describe the extremes of natural agents in wind engineering (Brabson and Palutikof, 2000; Gatey and Miller, 2007; Sacré et al., 2007; Torrielli et al., 2013; Valamanesh et al., 2015) and also in several other branches of civil engineering and geosciences. In line with this, a Poisson-Pareto model (Eq. 6) was used to characterize annual maximum values in this work.

$$Pr\left[X_{max} \leq x\right] = exp\left[-\nu\left(1 + \xi\frac{x-u}{\tilde{\sigma}}\right)^{-1/\xi}\right], \tag{6}$$





Where $\xi$, $\tilde{\sigma}$ and $u$ are, respectively, the parameters of form, scale, and location (threshold) of the Generalized Pareto Distribution (GPD), which is fit based on a POT regime (Hosking and Wallis, 1987); and where $\nu$ is the Poisson parameter which describes the annual mean rate of occurrence of these events.

To consider the displacement of the storms between sectors, the distribution of the annual maxima in a given sector $s$, was
calculated according to the next steps:

1. Storms were identified as the clusters of sequential values of the omnidirectional wind speed exceeding a given threshold, with a time lag of at least 5 days between their peaks to ensure their independence. In this way, the sequences of wind speeds and directions of the 270 omnidirectional storms were isolated.

2. From the set of 270 storms, the subset of those whose direction belongs at some point to the sector under consideration
was selected. The number of these storms was $n_s \leq 270$.

3. For each one of the $n_s$ storms, the maximum wind speed whose direction belongs to the sector $s$ was selected. These set of maximum values was the sample $m_s$.

4. A GPD with parameters $(\xi, \tilde{\sigma}, u)$ was fitted to the sample $m_s$ and the Poisson parameter $\nu$ was calculated.

5. The Poisson-Pareto model from Equation 6 was used for describing the distribution of the annual maxima in each sector.
From this distribution, any return level can be inferred.

### 3.4 Values adopted for the definition of requirements and indicators

The power curves of the Anderson-Darling test (Anderson and Darling, 1952) and the KS test (Kolmogoroff, 1941; Smirnoff, 1939) are necessary to define the minimum number of data in each sector and subsector. These curves were specifically obtained for this research by simulation (see the appendix for more details) for the usual values of $\beta = 0.2$ (probability of false
negative) and $\alpha = 0.05$ (significance level). The effect size was stated as the absolute displacement between the mean value of the population defining the null hypothesis and the mean value of the population from which the contrasted sample was extracted in each simulation. The value of the effect size was set at $0.5\sigma$, where $\sigma$ was the standard deviation of the reference population for the null hypothesis (Kottegoda and Rosso, 2008). According to this, sectors should contain a minimum of 50 data, and subsectors, a minimum of 15 data.

Furthermore, the study did not consider any division containing sectors that rejected the null hypothesis of the Anderson-Darling test (Anderson and Darling, 1952), with a significance level of $\alpha = 0.05$. To evaluate the $\overline{p_2}$ indicator, a reference return period of $T_r = 100$ years and admissible maximum discrepancy of $\pm\varepsilon_0 = 10\%$ in regard to the estimated value were used.

Finally, some practical limitations to the size of the sectors were considered for this case study in order to reduce the range
of divisions considered and to limit computational costs. Specifically, the sector amplitude was restricted to the range from $30°$ to $300°$ and only those sectors whose amplitude was a multiple of $5°$ were considered.





|  | $S_1$ | $S_2$ | $S_3$ | $S_4$ | $S_5$ | $S_6$ | $S_7$ | $S_8$ |
|---|---|---|---|---|---|---|---|---|
| Criterion T90 | 0-90 | 90-180 | 180-270 | 270-360 | - | - | - | - |
| Criterion T45 | 0-45 | 45-90 | 90-135 | 135-180 | 180-225 | 225-270 | 270-315 | 315-360 |
| Criterion C0 | 125-235 | 235-290 | 285-125 | - | - | - | - | - |
| Criterion C1 | 155-235 | 235-285 | 285-155 | - | - | - | - | - |
| Criterion C2 | 170-210 | 210-265 | 265-360 | - | - | - | - | - |
| Criterion C3 | 140-215 | 215-320 | 320-140 | - | - | - | - | - |

**Table 1.** Directional sectors corresponding to each criterion

### 3.5 Effect of the requirements and variation of indicators, depending on directional sectors

The effect of different criteria for sector selection on the extreme value models used to fit the available data was characterized. From now on, the following nomenclature is used to present the results: $C0$ is the criterion proposed in this work (whose definition is summarized in section 2.3) and $T90$ and $T45$ are the comparison criteria, which consist of sectors with a constant width of $90°$ and $45°$, respectively, and a northern direction of origin. Additionally, the definition of criteria $C1$, $C2$ and $C3$ also follows the procedure that is summarized in section 2.3, but at the point 4b of the aforementioned procedure, indicators $\overline{p_1}$, $\overline{p_2}$ and $\overline{p_3}$ are, respectively, used instead of the overall indicator $\|\overline{p_i}\|$.

When applying criteria $C0$, $C1$, $C2$ and $C3$, the selection requirements (section 2.1.1) reduce the number of candidate divisions to 15973. From these divisions, 58.5% correspond to three-sector divisions, 41.4% to four-sector divisions and 0.1% to five-sector divisions. The divisions resulting from each criterion are listed in Table 1. It should be highlighted that only sector $S_3$ of criterion $T90$ and sectors $S_5$ and $S_6$ of criterion $T45$ meet the pre-selection requirements imposed on the other ones ($C0$ to $C3$). This leads to a bias in the guarantees for modeling the directional extreme values given by both sets of criteria and should be kept in mind when judging their results.

Figures 4 and 5 show the characteristics of the extreme values in each division. Wind direction is represented on the x-axis, where north is zero, and wind magnitude is represented on the y-axis. Each graph presents the sectors that correspond to one of the criteria, as well as the boxplots, which show the median, the upper and lower quartiles, and the variability of the estimated 100 year return values in each sector. The sample of estimations was obtained by means of bootstrapping techniques. For this purpose, the omnidirectional storms were resampled with replacement. The sequence of speeds and directions of each storm remained fixed during each resampling and the size of the resample was always 270 (the original number of storms). Next, the directional 100 year return values from the resample were computed, and this routine was repeated 10000 times to get a precise estimate of the Bootstrap distribution of the statistic.

Also, for each criterion there is a scatter plot showing the data that was used for the fit of the directional extreme values. During any given storm, wind direction may vary in more than one sector and, in these cases, every storm produces more than one extreme value (one for each sector in which it has data). The number of points in each sector is indicated with the letter N.





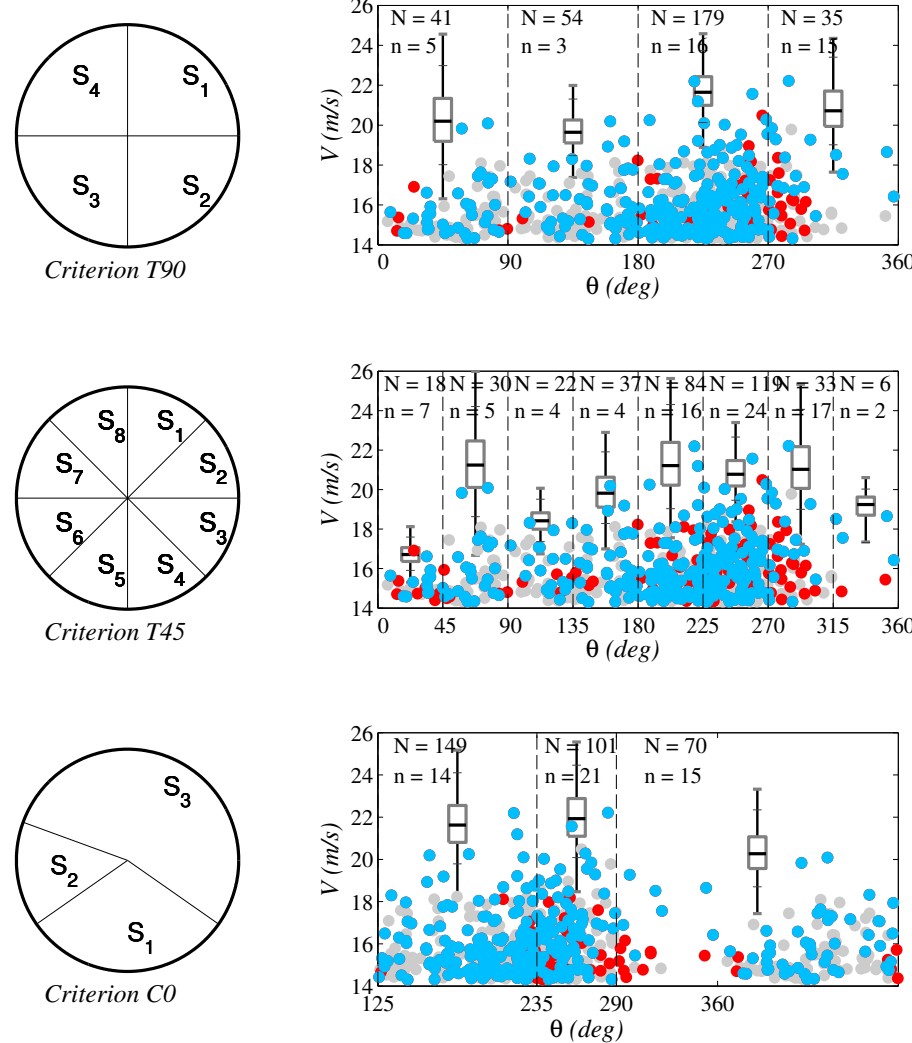

**Figure 4.** Sectors defined according to criteria T90, T45 and C0

These points are shown in colors blue and red. Blue points are the peak values of the omnidirectional storms. Hence, there are 270 blue points summed over all sectors. The red ones are the maxima in each sector of those storms whose omnidirectional peak occurred in a different sector. These points introduce dependency between the extremes of each sector, and the number of them is indicated with the letter n. Finally, threshold exceedances that do not take part on the fit are depicted in grey.

5     A comparison of the accuracy of the extreme value models that were used to fit the directional data of each criterion is shown in Figure 6. It depicts the quantile plots for the criteria $T90$ (row 1), $T450$ (rows 2 and 3) and $C0$ (row 4), where the x-axis corresponds to the empirical data and the y-axis to the models.




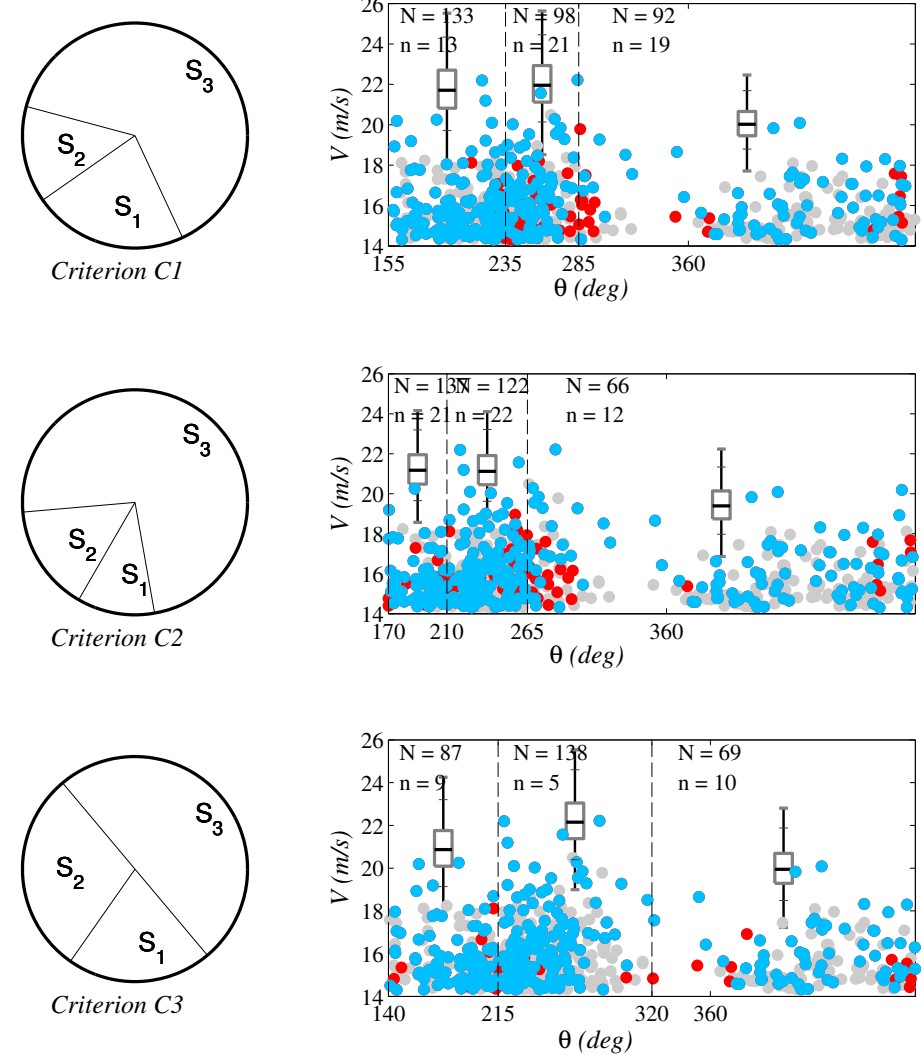

**Figure 5.** Sectors defined according to criteria $1-3$

Figure 7 measures the performance of each solution obtained in regard to indicators $\overline{p_1}$, $\overline{p_2}$ and $\overline{p_3}$. Each indicator is represented along its respective axis, which has its origin in the center. All axes are arranged radially (with equal distances from each other) and all of them have the same scale 0-1. To enhance the understanding of the graphs, the data is connected to form a polygon, and circles of iso-value are also represented. Table 2 shows the values for each indicator for the different criteria.

Criteria $C0$ to $C3$ lead to divisions with three sectors in all cases. A larger one, which roughly covers the W-SE region, and two more in the range where larger and more frequent storms occur. These divisions are consistent with the analysis of regional wind characteristics. Furthermore, the divisions of criteria $T90$ and $T45$ show worse results in all indicators with striking differences in that of intrasectorial homogeneity and independence of the sectorial extremes.





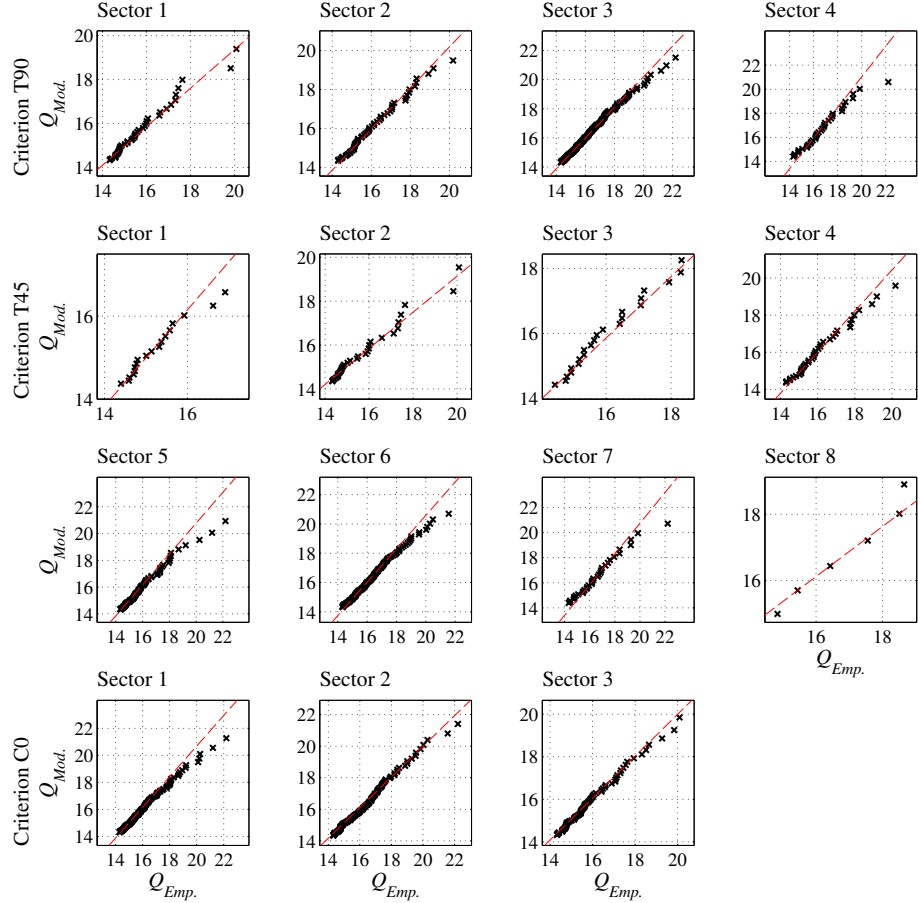

**Figure 6.** Quantile plots for fitted model in each sector (empirical quantile on the x-axis and model quantile on the y-axis). $T90$: row 1; $T45$: rows 2, 3; $C0$: row 4

|              | $\overline{p_1}$ | $\overline{p_2}$ | $\overline{p_3}$ |
|--------------|--------|--------|--------|
| Criterion T90 | 0.3150 | 0.9391 | 0.4571 |
| Criterion T45 | 0.4031 | 0.9266 | 0.2857 |
| Criterion C0 | 0.8485 | 0.9421 | 0.8527 |
| Criterion C1 | 0.9589 | 0.9512 | 0.6543 |
| Criterion C2 | 0.6111 | 0.9739 | 0.4790 |
| Criterion C3 | 0.4633 | 0.9440 | 0.9391 |

**Table 2.** Values of indicators for each criterion considered





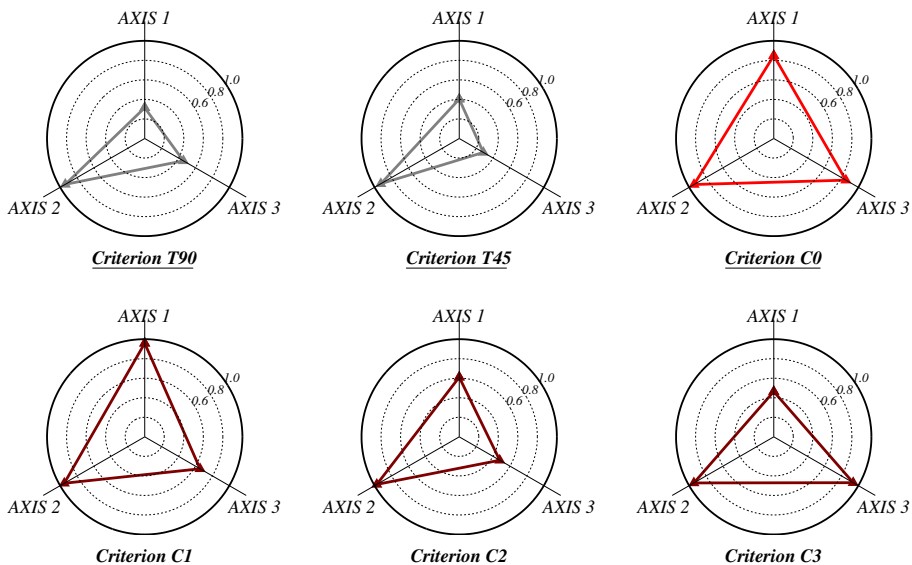

**Figure 7.** Indicators $\overline{p_1}$ (axis 1), $\overline{p_2}$ (axis 2), and $\overline{p_3}$ (axis 3) for each criterion considered

## 4   Dependence between sector selection and project design reliability

This section evaluates the effect of directional sector selection on design values and structure reliability. For this purpose, we used the simple example of a structure with three straight sections whose design wind speeds should be adapted to the directional variability of the extreme values of the agent. The normal directions of the sections form angles of 60°, 180°, and

300° in relation to the north. For ease of exposure, the following working hypotheses are assumed:

1. Failure of the whole structure occurs when at least one of the sections fails.

2. The failure mode does not depend on the direction of the agent's incidence but rather on the section type. In this case, the failure in each section occurs when wind action in the normal direction $\pm 22.5°$ exceeds a certain design value and is independent of the failure of the other sections.

3. The response coefficients of the structure are all equal to one.

4. Each directional sector isolates a population of the agent's extreme values with homogeneous characteristics.

Given the requirement that the overall failure probability in the useful life of the structure is lower than a given value $P_{f,V}$, there is an infinite number of compatible criteria that can define the failure probability of each section (Forristall, 2005). Jonathan and Ewans (2007) propose fixing these probabilities by minimizing the total cost $C$ of the structure, which they

define as an arbitrary function of the value of the design agent expressed as $C = K \sum_{n=1}^{N} x_n^2$, where $x_m$ is the design value in each section and $K$ is a constant. This research proposes an alternative function $C$ that incorporates two summands: (i) the



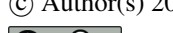

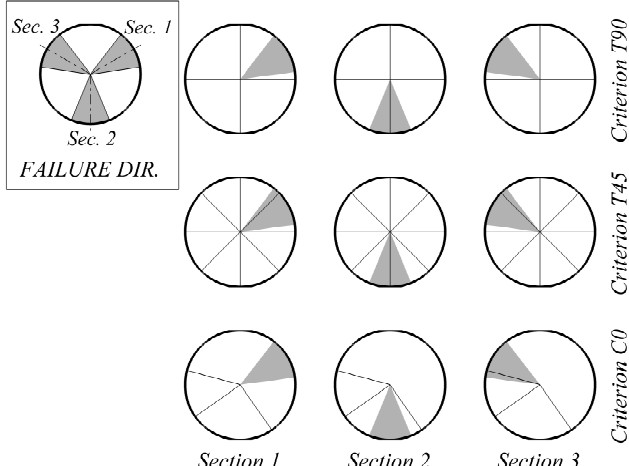

**Figure 8.** Failure regions in each section and their relation with the sectors of criteria $T90$ (upper row), $T45$ (middle row) and $C0$ (bottom row)

construction cost $C_c(x_n)$, which depends on the value of the design agent; (ii) the risk at each section obtained as the product of its probability of failure $P_{f_n,V}$ and its consequences $c_n$ (Losada, 2010) (Eq. 7).

$$C = \sum_{n=1}^{N} C_c(x_n) + \sum_{n=1}^{N} c_n P_{f_n,V} \tag{7}$$

We compared the design obtained with the sectors defined according to criterion $C0$ with the result of applying (a) the
5 omnidirectional analysis (where the design value of the wind speed is the same for the three sections) and (b) the sectorial divisions obtained from criteria $T90$ and $T45$. For each section and criterion, Figure 8 shows the range of directions that can cause failure (in gray) as well as the directional sectors involved.

Assuming a section n, which is affected by a number $S$ of directional sectors, the probability of an annual failure $P_{f_n,1}$ is obtained with Eq. 8. This equation expresses the complementary value that the annual maximum of the agent not exceed the
10 design value $x_n$ in any of the subsectors s (which are assumed to be independent) and where $\nu_s$, is the Poisson parameter, and $\xi_s$ and $\widetilde{\sigma}_s$ are, respectively, the form and scale parameters of the fitted generalized Pareto distribution in this subsector.

$$P_{f_n,1} = 1 - \prod_{s=1}^{S} Pr\left[ X_{\underset{1}{max,s}} \leq x_n \right] = 1 - \prod_{s=1}^{S} exp\left[ -\nu_s \left( 1 + \xi_s \frac{x_n - u}{\widetilde{\sigma}_s} \right)^{-1/\xi_s} \right] \tag{8}$$





For a maximum admissible failure probability of $p_0$ and a useful life of $V$ years (Eq. 9), the following optimization problem was formulated:

$$\min \quad C = C(x_n, c_n) \quad \forall n \in \mathbb{N}[1, N]$$

subject to:

$$(9)$$

$$P_{f_n,1} \leq 1 - (1 - p_0)^{1/V}$$

$$x_n = \max_n \{x_s\}$$

$$x_n \in \mathbb{R}^+ \quad \forall n \in \mathbb{N}[1, N]$$

Where $N$ is the number of sections; $P_{f_n,1}$ is provided by Eq. 8; and $x_s$ is the design value in each of the subsectors that

affect the section. For the objective function C, we adopted the expression given by Eq. 10. In this expression, $K = 0.025$ such that the design cost for a wind speed of 20 m/s is 10. The value of the consequences is related to the admissible maximum failure probability in the useful life of the structure (Losada, 2002). Accordingly, $c_n = 50$ was adopted for $p_0 \leq 0.2$ (very low repercussions) and $c_n = 100$ for $p_0 \leq 0.1$ (low repercussions), such the risk is 10 in both cases.

$$C = \sum_{n=1}^{N} K x_n^2 + \sum_{n=1}^{N} c_n P_{f_n,V} \tag{10}$$

## 4.1 Extreme value models

For the omnidirectional analysis we fit a Generalized Pareto Distribution (GPD) to the omnidirectional POT regimen. Next, we used a Poisson-Pareto model to estimate the distribution for the annual maxima in the range of directions that can cause failure of each sector (the Poisson parameter was evaluated into these ranges). For the directional results we followed the same scheme but the GPD was fit to the directional POT regimes according to the divisions of each criterion.

Table 3 shows the characteristics of the extreme-value models that intervene in the optimization problem of each criterion. The differences between criteria can be seen, for example, in the upper bound for wind speed in each section, which is limited by $u - \widetilde{\sigma}_s / \xi_s$ if $\xi_s \leq 0$. The greater discrepancies can be found in Section 1, where this bound has values 48.6 m/s and 24.4 m/s for criteria $T90$ and $C0$, and it does not exist for $T45$. For measuring how well the GPD fits the input data, the last two columns show, respectively, the statistic $K$ and the critical value $CV$ of the KS goodness of fit test with a significance level of

0.05. In all cases, the test statistic is less than the critical value.

## 4.2 Optimization and design wind speeds

The optimization problem was solved with an interior point algorithm (Byrd et al., 2000). Table 4 shows the design wind speeds obtained in each section, depending on the criterion ($C0$, $T90$, $T45$ and omnidirectional). The upper rows of the table



| Criterion T90 | Section | $\varphi_s$ | $\nu_s$ | $\xi_s$ | $\widetilde{\sigma_s}$ | $u$ | $K$ | $CV$ |
|---|---|---|---|---|---|---|---|---|
| | | deg | [-] | [-] | [m] | [m] | [-] | [-] |
| *Sector 1* | 1 | 45 | 0.554 | -0.043 | 1.482 | 14.3 | 0.0978 | 0.2076 |
| *Sector 2* | 2 | 22.5 | 0.365 | -0.441 | 2.724 | 14.3 | 0.0696 | 0.1814 |
| *Sector 3* | 2 | 22.5 | 1.210 | -0.260 | 2.439 | 14.3 | 0.0339 | 0.1005 |
| *Sector 4* | 3 | 45 | 0.473 | -0.504 | 3.680 | 14.3 | 0.0879 | 0.2243 |
| Criterion T45 | Section | $\varphi_s$ | $\nu_s$ | $\xi_s$ | $\widetilde{\sigma_s}$ | $u$ | $K$ | $CV$ |
| *Sector 1* | 1 | 7.5 | 0.081 | -0.466 | 1.376 | 14.3 | 0.1413 | 0.3094 |
| *Sector 2* | 1 | 37.5 | 0.676 | 0.110 | 1.340 | 14.3 | 0.0969 | 0.2417 |
| *Sector 4* | 2 | 22.5 | 0.500 | -0.353 | 2.542 | 14.3 | 0.0980 | 0.2183 |
| *Sector 5* | 2 | 22.5 | 1.135 | -0.137 | 1.880 | 14.3 | 0.0622 | 0.1461 |
| *Sector 7* | 3 | 37.5 | 0.743 | -0.351 | 3.109 | 14.3 | 0.0844 | 0.2308 |
| *Sector 8* | 3 | 7.5 | 0.027 | -0.750 | 4.560 | 14.3 | 0.1865 | 0.5193 |
| Criterion C0 | Section | $\varphi_s$ | $\nu_s$ | $\xi_s$ | $\widetilde{\sigma_s}$ | $u$ | $K$ | $CV$ |
| *Sector 1* | 1 | 45 | 0.437 | -0.221 | 1.9978 | 14.3 | 0.0599 | 0.1598 |
| *Sector 2* | 2 | 45 | 1.647 | -0.157 | 1.930 | 14.3 | 0.0446 | 0.1101 |
| *Sector 3* | 3 | 12.5 | 0.620 | -0.255 | 2.622 | 14.3 | 0.0474 | 0.1334 |
| *Sector 1* | 3 | 32.5 | 0.315 | -0.221 | 1.997 | 14.3 | 0.0599 | 0.1598 |

**Table 3.** Parameters of the optimization problem for criteria $T90$, $T45$, and $C0$

| | Vel. [m/s] | $T90$ | $T45$ | $C0$ | *Omni.* |
|---|---|---|---|---|---|
| $p_0 \leq 0.2$ | *Section 1* | 23.21 | 26.01 | 21.79 | 23.08 |
| | *Section 2* | 22.86 | 23.61 | 23.55 | 23.08 |
| | *Section 3* | 21.59 | 22.85 | 23.3623 | 23.08 |
| $p_0 \leq 0.1$ | *Section 1* | 23.98 | 27.63 | 22.06 | 23.32 |
| | *Section 2* | 23.04 | 24.06 | 23.92 | 23.32 |
| | *Section 3* | 21.60 | 22.95 | 23.62 | 23.32 |

**Table 4.** Optimization results for criteria $T90$, $T45$, $C0$ and omnidirectional analysis for $p_0 = 0.2$ (top rows) and $p_0 = 0.1$ (bottom rows)

show the results for an admissible maximum failure probability in $V = 50$ years of $p_0 \leq 0.2$, whereas the lower rows show the results for $p_0 \leq 0.1$.

All the comparison criteria ($T90$, $T45$ and omnidirectional) show design wind speeds for each section that differ from those of criterion $C0$. Consequently, directional sectors selection can be decisive in the project design if cost is a relevant factor. The greatest discrepancies occurred in section 1. With criterion $T90$, there are variations of 6.54% with $p_0 \leq 0.2$ and of 8.67% with





|  |  | T90 | T45 | C0 | Omni. |
|---|---|---|---|---|---|
| $p \leq 0.2$ | $P_f$ | 0.2584 | 0.0348 | 0.0260 | 0.0445 |
|  | $C$ | 51.5713 | 45.8301 | 40.6956 | 42.1913 |
| $p \leq 0.1$ | $P_f$ | 0.2473 | 0.0265 | 0.0111 | 0.0263 |
|  | $C$ | 64.7100 | 49.3822 | 41.5331 | 43.4366 |

**Table 5.** Failure probabilities and cost function (Eq. 10) of $T90$, $T45$, $C0$ and omnidirectional criteria for $p_0 \leq 0.2$ (top row) and $p_0 \leq 0.1$ (bottom row)

$p_0 \leq 0.1$. With criterion $T45$, there are variations of 19.36% for $p_0 \leq 0.2$ and of 25.24% for $p_0 \leq 0.1$. With the omnidirectional criterion, variations are 5.91% and 5.70%, respectively.

By definition, the design wind speeds corresponding to each criterion fulfill the requirements of the optimization problem, in accordance with their respective probability models. However, only the sectors of criterion C0 have been selected objectively in consonance with the working hypothesis of the directional model for the extremes and, therefore, they offer better guarantees for dimensioning. Thus, in order to compare the impact of ñdesign wind speeds on both, the failure probability during the useful life of the structure and the cost function (Eq. 10), the extreme-values model corresponding to $C0$ was used as a reference.

Table 5 indicates the total failure probability in the useful life of the structure and the result of the cost function. These values were calculated by entering the design wind speeds of each criterion in the directional model obtained from $C0$. The first rows show the results for $p_0 \leq 0.2$, and the last ones show those for $p_0 \leq 0.1$.

Solving the optimization problem for criterion $C0$ led to solutions far from the edge of the validity region. The design wind speeds obtained with the other criteria increase the probability of failure but fulfill the design requirements, with the exception of $T90$ criterion. Particularly noteworthy is the result with $T90$ for $p_0 \leq 0.1$, which almost doubles the maximum acceptable probability of failure during useful life. Regarding the cost function, notable differences can be found, with increases by 26.7%, 12.6% and 3.7% for $T90$, $T45$ and omnidirectional criterion, respectively for $p_0 \leq 0.2$, and 55.8%, 18.9% and 4.6% for $p_0 \leq 0.1$. These differences show that the selection of directional sectors can have significant implications for the calculation of structure reliability and costs and, thus, should be included as an integral part of project design.

On a final note, the selection of the threshold is an additional source of uncertainty which can affect the results. Preliminary analysis suggest that the calculation of reliability may be sensitive to the directional variability of this threshold. Nevertheless, a deeper study is still needed to properly incorporate the effect of this variability on the definition of homogeneous and independent sectors and its impact on the uncertainty of the results.

# 5 Conclusions

This paper has described a procedure for the selection of directional sectors in a non-arbitrary manner, considering the following sources of uncertainty: (1) the validity of the model used to characterize the extreme behavior of the sector samples; (2) the





goodness of parameter estimation; (3) the capacity of each model to represent extreme behavior in the total amplitude of the corresponding sector; (4) the validity of the working hypothesis of the independence between extreme values in different sectors.

This research led to the following conclusions. Firstly, the results of modeling the directional extreme behavior of natural agents can be affected by the choice of directional sectors used for calculation. Secondly, the selection of sectors without considering the extreme properties of the data negatively affects the confidence in the estimates on which the project design is based. This makes the use of sectors of equal amplitude not recommended, without sufficient justification. In this sense, the method presented in this research is an objective tool for the selection of directional sectors, which also facilitates the application of standard calculation procedures since it leads to homogeneous and uncorrelated sectors. The results obtained show that it offers better guarantees for dimensioning than the use of more conventional engineering approaches based on divisions arbitrarily chosen, because it reduces the sources of uncertainty in the estimation of design values. Furthermore, this method also assures that sector division by direction is in consonance with the working hypotheses of the directional model. This means that quantification of probabilities is applied within the validity range of this model.

The method was applied to the selection of directional sectors for the calculation of the design wind speed of a structure located at the mouth of the Río de la Plata. The impact that choice of method would have on the failure probability during the useful life of a structure was analyzed, and the results with the proposed method were compared to those based on divisions with equal size sectors and a northern direction of origin. It was found that the procedure followed can have significant repercussions on the cost estimate and reliability, and thus condition the viability of an investment project. Consequently, decisions regarding sector selection should be an integral part of the project design process.

**Appendix A**

The procedure for obtaining the power curves for the Anderson-Darling test, whose null hypothesis, $H_0$, is that the sample belongs to a population with a distribution function, $F(x) \sim GPD(\xi, \tilde{\sigma}, u)$ with mean $\mu$ and standard deviation $\sigma$, was the following:

1. Select the significance level $\alpha$ and effect size $c$.

2. Define the parameters of a generalized Pareto distribution $G(x) \sim GPD(\xi^*, \tilde{\sigma}^*, u)$ with mean value $\mu^* = \mu + c$ and standard deviation $\sigma$.

3. Simulate a number N of random samples with a distribution function $G(x)$ for each sample size of interest.

4. Obtain the result (rejection/non-rejection ) of the Anderson-Darling test for each simulated sample.

5. Obtain the value of $\beta$ for each size as the quotient of the number of positive results and the total number of simulations.

Similarly, the power curves for a Kolmogoroff-Smirnoff test whose null hypothesis, $H_0$, is that two samples belong the same population, can be obtained as follows:





1. Select the significance level $\alpha$ of the effect size $c$ and of a sample size $M$.

2. Define the parameters for two generalized Pareto distributions: (i) $F(x) \sim GPD(\xi, \tilde{\sigma}, u)$ with mean $\mu$ and standard deviation $\sigma$,; (ii) $G(x) \sim GPD(\xi^*, \tilde{\sigma}^*, u)$ with mean $\mu^* = \mu + c$ and standard deviation.

3. Simulate a number N of random samples with distribution function $F(x)$ and $G(x)$ for each value of $\tilde{m} = min|m, m'|$ of interest, where $m$ and $m'$ are, respectively, the sample sizes of $F(x)$ and $G(x)$, and $m + m' = M$.

4. Obtain the result (rejection/non-rejection) of the Kolmogoroff-Smirnoff test for each pair of simulated samples.

5. Obtain the value of $\beta$ for each size $\tilde{m}$ as the quotient of the number of positive results and the total number of simulations.

*Competing interests.* The authors declare that they have no conflict of interest.





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
