# Peer review of "The selection of directional sectors for the analysis of extreme wind speed"

_Natural Hazards and Earth System Sciences, 2018_

## Referee Comment (RC1) · Anonymous Referee #1 · 1 Sep 2018

This paper describes a method for finding the optimum set of directional sectors for a directional extreme value analysis. The goal is interesting and the proposed method seems plausible, but it failed to convince me that the results are optimal. In the example, the sections for analysis are separated from each other. Why are the proposed sectors superior to sectors that match the sections? The sections are far enough separated that independence should not be an issue. The intuitive choice would be to pick sectors centered on the sections and as wide as the data appears to be homogeneous. Why is that not better?

According to Table 2 and Figure 7, all of the extreme value fits are good and do not affect the sector choices. But some of the fits in Figure 6 do not appear to be all that good. (Incidentally, I applaud the selection of Figures that give details of the process).

[Figure]

The fit for Sector 1 in C45 is considerably higher than the empirical data. Is that related to why the extreme wind speed for Section 1 and T45 is so high? By comparison with results in Figure 4, 26 m/sec does not seem reasonable.

There are a few places where the text is not clear and I had to read farther on through the examples to understand the process.

In heading 3.1, what does "agent" at the site mean?

On line 7, page 8, what are "two moments"? Are they six hours apart or do they include the whole storm?

On lines 12-14, page 8, are the peak events just the peak Hs in each storm or the peak Hs in each sector in each storm?

The caption for Table 1 would be much clearer if it read "Directional sectors resulting from applying the different selection criteria."

In equation (8), I don't see where the width of the sector appears.

———————————————————

---

## Referee Comment (RC2) · Anonymous Referee #2 · 30 Oct 2018

The paper present an original method to be employed fro the selection of directional sectors for the analysis of the extreme wind speed in order to develop design of structures exposed to wind action. I found the subject of the manuscript relevant for NHESS and I think that it could be of interest for other scientific and engineering field such as, for example, coastal and offshore engineering. The formulation of the problem, the presentation and the discussion of the results are clear and adequately extended. I have some comment and observation about some aspect of the analysis presented by the authors: a) The authors decided to employ a criterion for the selection of directional sectors based on different statistical requirements and indicators. What could be the difference by the use with some sort of clustering techniques (k-means for example, or similar)? Could the authors comment on this point and eventually add some discussion

in the paper b) I would make dimensionless the global indicator: instead of varying be-tween 0 and sqrt(3) I would make it varying between 0 and 1 c) I would add some plot about the minimum data for sectors and subsectors (first paragraph of section 3.4) d) I would put the x scale of figure 4 for criterion C0 varying from 0 to 360 in order to have a visual comparison with Criterion 45 and 90 e) May be I miss something but I do not understand why sigma_s and u are in [m] in table 3 f) I would add bound conditions in table 3 in order to have a clear defined picture of the quantities involved g) There is any effect on the results on the choice of different inter arrival time (different from 5 days)? h) It is not clear to me how the threshold has been chosen (and it would be nice to have indication about changes in the results depending on the threshold) i) in last line of page 7 there is a typo "rfrg"

---

## Author Comment (AC2) · 19 Nov 2018

***Authors' response to comments by referee #2***

*Manuscript title: The selection of directional sectors for the analysis of extreme wind speed. We would like to thank the referee for all the valuable comments that will help us to improve our manuscript and future research. Comments from the referee are written in italics, followed by our response.*

**Referee #2**

*The paper presents an original method to be employed for the selection of directional sectors for the analysis of the extreme wind speed in order to develop design of structures exposed to wind action. I found the subject of the manuscript relevant for NHESS and I think that it could be of interest for other scientific and engineering field such as, for example, coastal and offshore engineering. The formulation of the problem, the presentation and the discussion of the results are clear and adequately extended. I have some comment and observation about some aspect of the analysis presented by the authors:*

a) *The authors decided to employ a criterion for the selection of directional sectors based on different statistical requirements and indicators. What could be the difference by the use with some sort of clustering techniques (k-means for example, or similar)? Could the authors comment on this point and eventually add some discussion in the paper*

   Starting from the information of wind direction and speed, a cluster analysis methodology would allow to determine subsets of similar data in terms of some distance measure that could be defined in terms of these two variables. This can be seen as a way to define directional sectors for the extreme analysis. However, cluster analysis does not usually include metrics to ensure that data subsets are homogeneous and independent of each other. In turn, the use of cluster analysis requires defining the clustering methodology, the distance measure between data and the number of clusters.

   It is possible that the methodological approach suggested in this article can be adapted to guide the analyst in the definition of these three points in order to obtain directional groups that meet the requirements imposed for the extremal analysis (i.e. homogeneous and independent sectors such that the variance of estimated extremes is minimized).

b) *I would make dimensionless the global indicator: instead of varying between 0 and sqrt(3) I would make it varying between 0 and 1*

   We agree with the referee that varying the global indicator between 0 and 1 leads to a more direct understanding of its value and it does not change the method or the results in any way. For that reason, we have changed equation 5 according to the referee's proposal.

c) *I would add some plot about the minimum data for sectors and subsectors (first paragraph of section 3.4)*

   The power curves for the Anderson Darling and Kolmogoroff-Smirnoff tests have been obtained as indicated in Appendix A and have been added to the manuscript (section 3.4), along with a more detailed description of their use in this article.

*d) I would put the x scale of figure 4 for criterion C0 varying from 0 to 360 in order to have a visual comparison with Criterion 45 and 90*

Varying the x scale of C0 from 0 to 360 divides sector S3 into two parts (one on the right and the other on the left of the figure). In our opinion, this makes it more difficult to understand the figure and to place the boxplots in a consistent manner. For this reason, although the referee's proposal has some advantages, we believe it is better to maintain the figure 4 in its current state.

*e) May be I miss something but I do not understand why sigma_s and u are in [m] in table 3*

We appreciate the referee's comment. The units for "sigma_s" and "u" are [m/s]. We have changed the units in columns 6 and 7 in table 3 accordingly.

*f) I would add bound conditions in table 3 in order to have a clear defined picture of the quantities involved*

We agree with the referee that the inclusion in table 3 of a column with the upper bound of the wind velocity in each sector helps to illustrate the differences between the extreme-value models for each criterion, as indicated in section 4.1.

*g) There is any effect on the results on the choice of different inter arrival time (different from 5 days)?*

The inter arrival time between independent storms should have a physical/statistical sense and can be chosen either by rational methods or experience. In any case, performing a sensitivity analysis for testing the impact of this parameter on the results is recommended.

In the case study, an inter arrival time of 5 days between the peaks of consecutive storms has been adopted, which leads to 270 storms. Table 1 shows the results for inter arrival times of 2, 5, 7 and 10 days. In all these cases, the number of directional sectors obtained according to the method is three. First column shows the inter arrival time; second one the number of storms; from third to fifth column, the directional sectors; and from sixth to eighth column, the 100 year return value in each sector.

| $\Delta t$ | $N_s$ | $S_1$ | $S_2$ | $S_3$ | $u_{100,S1}$ | $u_{100,S2}$ | $u_{100,S3}$ |
|---|---|---|---|---|---|---|---|
| [days] | [-] | [deg] | [deg] | [deg] | [m/s] | [m/s] | [m/s] |
| 1 | 294 | 140-235 | | 290-140 | 21.8 | 22.1 | 20.4 |
| 2 | 288 | | | | 21.7 | 22.2 | 20.4 |
| **5** | **270** | **125-235** | **235-290** | **290-125** | **21.8** | **22.1** | **20.5** |
| 7 | 256 | | | | 21.6 | 21.9 | 20.7 |
| 10 | 238 | | | | 21.4 | 21.8 | 20.3 |

TABLE 1

Results show how, in the case study, the method is very little sensitive to changes in the inter arrival time. The directional sectors are the same for the inter arrival times of 2, 5, 7 and 10 days and differ in 15 degrees for the inter arrival time of 1 day. The 100 year return values are also consistent, with variations of less than 2.1% in all sectors.

*h) It is not clear to me how the threshold has been chosen (and it would be nice to have indication about changes in the results depending on the threshold)*

In this study, the threshold used to define the extreme data was chosen by applying the method in Solari et al. (2017), i.e. the threshold that maximizes the p-value of the Anderson-Darling test, under the assumption that the omnidirectional data come from a Generalized Pareto Distribution.

As shown in Table 2, directional sectors are sensitive to changes in the threshold u (first column), i.e., different definitions of what is an extreme value result in different directional sectors. However, for thresholds above the chosen one, the sectors are quite stable (in particular S2). Therefore, if there is no strong criterion for an a priori selection of the threshold, a sensitivity analysis of the results is recommended. In this situation, $p_0$ (now defined as suggested in (b) in column 9) could serve as an indicator for the final choice of the threshold.

These considerations have been included in the last paragraph of section 4.2.

| u | $N_s$ | $S_1$ | $S_2$ | $S_3$ | $u_{100,S1}$ | $u_{100,S2}$ | $u_{100,S3}$ | $p_0$ |
|---|---|---|---|---|---|---|---|---|
| [m/s] | [-] | [deg] | [deg] | [deg] | [m/s] | [m/s] | [m/s] | [-] |
| 14.0 | 306 | 20-140 | 140-195 | 195-20 | 19.9 | 21.5 | 23.0 | 0.9489 |
| **14.3** | **270** | **125-235** | **235-290** | **290-125** | **21.8** | **22.1** | **20.5** | **0.8822** |
| 14.5 | 251 | 145-235 | 235-285 | 285-145 | 22.5 | 22.0 | 20.5 | 0.8799 |
| 15.0 | 203 | 200-235 | 235-290 | 290-200 | 23.0 | 20.7 | 21.5 | 0.8668 |

TABLE 2

Solari, S., Egüen, M., Polo, M. J., & Losada, M. A. (2017). Peaks Over Threshold (POT): A methodology for automatic threshold estimation using goodness of fit p-value. Water Resources Research, 53(4), 2833-2849.

*i) in last line of page 7 there is a typo "rfrg"*

We have corrected this typo.

---

## Author Response (AR1)

In these Revision Notes we explain the changes and corrections we have made to the manuscript according to all reviewers' comments. The document is structured following the sequence: (1) comments from reviewers, (2) authors' response, and (3) authors' changes in manuscript.

At the end of these notes, the revised manuscript showing the new text using highlighting can be found.

**REVIEWER #1**

*This paper describes a method for finding the optimum set of directional sectors for a directional extreme value analysis.*

1. *The goal is interesting and the proposed method seems plausible, but it failed to convince me that the results are optimal. In the example, the sections for analysis are separated from each other. Why are the proposed sectors superior to sectors that match the sections? The sections are far enough separated that independence should not be an issue. The intuitive choice would be to pick sectors centered on the sections and as wide as the data appears to be homogeneous. Why is that not better?*

   The reviewer says he is not convinced that the proposed method is optimal. However, the authors do not pretend to be presenting an "optimal" method, but an objective method of directional classification for directional extreme value analysis. The current state of the art on selection of sectors for directional analysis is subjective. In this work, the objective is to propose an objective method "considering the main sources of uncertainty stemming from sector selection: (1) the validity of the model used to characterize the extreme behavior of the sector samples; (2) the goodness of parameter estimation; (3) the capacity of each model to represent extreme behavior in the total amplitude of the corresponding sector; (4) the validity of the working hypothesis of the independence between extreme values in different sectors" (quoted from first paragraph of section 5).

   Whether or not the sectors division is optimal will depend on its application and on the objective function and the constraints of the optimization. This analysis is out of the scope of this manuscript.

   What the authors do intend is the proposed method to be general and not conditioned by the subsequent use made of the directional sectors, e.g. in the case of the example presented in the manuscript, that the method is not conditioned by the structure that is being designed. In this sense, it is possible that, as the reviewer points out, other specific directional partitioning methodologies can be defined for the analysis of the structure in question, and that these can be considered "optimal" for that particular case. It is important to note that the methodological approach and the tools proposed in this paper would also be useful when objectively determining the sectors if the approach proposed by the reviewer is followed (i.e. conditioned to the structure being designed).

   CHANGES TO THE MANUSCRIPT

   We believe that authors' intention with the proposed method is already present in the manuscript (see, e.g., the fourth and fifth paragraphs of the introduction or the first two paragraphs of the conclusions) and, therefore, we have not added any additional comment in this regard.

2. *According to Table 2 and Figure 7, all of the extreme value fits are good and do not affect the sector choices. But some of the fits in Figure 6 do not appear to be all that good. (Incidentally, I applaud the selection of Figures that give details of the process). The fit for Sector 1 in C45 is considerably higher than the empirical data. Is that related to why the extreme wind speed for Section 1 and T45 is so high? By comparison with results in Figure 4, 26 m/sec does not seem reasonable.*

The QQ graph for sector 1 with criterion T45 has values between 14 m/s and approx. 17 m/s. It shows that the model slightly underestimates the observations (in less than 0.5 m/s), as opposed to what was indicated by the reviewer.

The large values obtained for section 1 in the case of criterion T45 (we assume that the reviewer refers to table 4) are likely to come from sector 2, where the fit of the GPD results in a positive shape parameter (approx. 0.1), unlike all others fits, where the shape parameter is always negative (see table 3). This, if you will, highlights once again the drawbacks of using arbitrary sectors for the analysis of directional extremes.

CHANGES TO THE MANUSCRIPT

We have made no changes in relation with this question in the manuscript.

3. *There are a few places where the text is not clear and I had to read farther on through the examples to understand the process. In heading 3.1, what does "agent" at the site mean?*

Given that the proposed methodology is applicable not only to wind, but also to other directional climatic agents, such as waves or currents, it was decided to use "agents" instead of "wind" in many paragraphs throughout the text to highlight this generality.

CHANGES TO THE MANUSCRIPT

In the introduction (line 14, page 2), we have specified some examples of "climatic agents" to set this concept clear from the beginning. We have also changed the heading 3.1 (line 21, page 6) in order to highlight the agent that was analysed in the case study.

4. *On line 7, page 8, what are "two moments"? Are they six hours apart or do they include the whole storm?*

The whole storm is included. It refers to the absolute maximum difference of the wind direction that can be found between any two points in time within a given storm, taking into account the direction of rotation (clockwise or counterclockwise).

CHANGES TO THE MANUSCRIPT

To improve the clarity of the text, we have changed the previous explanation to the one above (lines 6-8, page 8).

5. *On lines 12-14, page 8, are the peak events just the peak Hs in each storm or the peak Hs in each sector in each storm?*

For every storm, we obtain the peak of the wind speed for each one of the sectors that the storm passes through.

CHANGES TO THE MANUSCRIPT

We have replaced the previous explanation with the one above (lines 13-14, page 8).

6. **The caption for Table 1 would be much clearer if it read "Directional sectors resulting from applying the different selection criteria."**

We appreciate the reviewer's suggestion and we will include it in the new version of the manuscript.

CHANGES TO THE MANUSCRIPT

We have changed the caption for Table 1 according to the reviewer's suggestion (page 11).

7. **In equation (8), I don't see where the width of the sector appears.**

The width of the subsector is considered in the calculation of each Poisson parameter nu_s. This parameter indicates the annual rate of peaks within the subsector, therefore it is influenced by subsector's width.

CHANGES TO THE MANUSCRIPT

We have improved the explanation with a specific mention to the sectors' width (lines 4-5, page 17).

**REVIEWER #2**

*The paper presents an original method to be employed for the selection of directional sectors for the analysis of the extreme wind speed in order to develop design of structures exposed to wind action. I found the subject of the manuscript relevant for NHESS and I think that it could be of interest for other scientific and engineering field such as, for example, coastal and offshore engineering. The formulation of the problem, the presentation and the discussion of the results are clear and adequately extended. I have some comment and observation about some aspect of the analysis presented by the authors:*

1. **The authors decided to employ a criterion for the selection of directional sectors based on different statistical requirements and indicators. What could be the difference by the use with some sort of clustering techniques (k-means for example, or similar)? Could the authors comment on this point and eventually add some discussion in the paper**

Starting from the information of wind direction and speed, a cluster analysis methodology would allow to determine subsets of similar data in terms of some distance measure that could be defined in terms of these two variables. This can be seen as a way to define directional sectors for the extreme analysis. However, cluster analysis does not usually include metrics to ensure that data subsets are homogeneous and independent of each other. In turn, the use of cluster analysis requires defining the clustering methodology, the distance measure between data and the number of clusters.

It is possible that the methodological approach suggested in this article can be adapted to guide the analyst in the definition of these three points in order to obtain directional groups that meet the requirements imposed for the extremal analysis (i.e. homogeneous and independent sectors such that the variance of estimated extremes is minimized).

CHANGES TO THE MANUSCRIPT

A formal discussion on this interesting topic would need some introduction and specific details about clustering techniques, which is, in the authors' opinion, outside the scope of this article. Therefore, we believe that the above considerations fit better in the interactive discussion in NHESSD.

2. *I would make dimensionless the global indicator: instead of varying between 0 and sqrt(3) I would make it varying between 0 and 1*

We agree with the referee that varying the global indicator between 0 and 1 leads to a more direct understanding of its value and it does not change the method or the results in any way.

CHANGES TO THE MANUSCRIPT

We have changed Equation 5 (page 6) according to the referee's proposal, as well as the explanatory text (lines 8-9, page 6).

3. *I would add some plot about the minimum data for sectors and subsectors (first paragraph of section 3.4)*

The power curves for the Anderson Darling and Kolmogoroff-Smirnoff tests have been obtained as indicated in Appendix A and have been added to the manuscript (section 3.4), along with a more detailed description of their use in this article.

CHANGES TO THE MANUSCRIPT

Three new plots (Figure 4, page 10) have been added to the manuscript and the first paragraph of section 3.4 (lines 20-29, page 9) has been completely rewritten.

4. *I would put the x scale of figure 4 for criterion C0 varying from 0 to 360 in order to have a visual comparison with Criterion 45 and 90*

Varying the x scale of C0 from 0 to 360 divides sector S3 into two parts (one on the right and the other on the left of the figure). In our opinion, this makes it more difficult to understand the figure and to place the boxplots in a consistent manner. For this reason, although the referee's proposal has some advantages, we believe it is better to maintain the figure 4 in its current state.

CHANGES TO THE MANUSCRIPT

We have not made any changes related to this question in the manuscript.

5. *May be I miss something but I do not understand why sigma_s and u are in [m] in table 3*

We appreciate the referee's comment. The units for "sigma_s" and "u" are [m/s].

CHANGES TO THE MANUSCRIPT

We have corrected the units in columns 6 and 7 in Table 3 (page 17).

6. *I would add bound conditions in table 3 in order to have a clear defined picture of the quantities involved*

We agree with the referee that the inclusion in Table 3 of a column with the upper bound of the wind velocity in each sector helps to illustrate the differences between the extreme-value models for each criterion, as indicated in section 4.1.

CHANGES TO THE MANUSCRIPT

We have included a new column in Table 3 with the upper bound of the wind velocity in each sector (page 17), as well as a reference to the nomenclature (line 8, page 17). We have also corrected an error in the bound value for criterion C0 in the text (lines 9-10, page 17).

**7. There is any effect on the results on the choice of different inter arrival time (different from 5 days)?**

The inter arrival time between independent storms should have a physical/statistical sense and can be chosen either by rational methods or experience. In any case, performing a sensitivity analysis for testing the impact of this parameter on the results is recommended.

In the case study, an inter arrival time of 5 days between the peaks of consecutive storms has been adopted, which leads to 270 storms. Table 1 shows the results for inter arrival times of 2, 5, 7 and 10 days. In all these cases, the number of directional sectors obtained according to the method is three. First column shows the inter arrival time; second one the number of storms; from third to fifth column, the directional sectors; and from sixth to eighth column, the 100 year return value in each sector.

| $\Delta t$ | $N_s$ | $S_1$ | $S_2$ | $S_3$ | $u_{100,S1}$ | $u_{100,S2}$ | $u_{100,S3}$ |
|---|---|---|---|---|---|---|---|
| [days] | [-] | [deg] | [deg] | [deg] | [m/s] | [m/s] | [m/s] |
| 1 | 294 | 140-235 | | 290-140 | 21.8 | 22.1 | 20.4 |
| 2 | 288 | | 235-290 | | 21.7 | 22.2 | 20.4 |
| **5** | **270** | 125-235 | | 290-125 | **21.8** | **22.1** | **20.5** |
| 7 | 256 | | | | 21.6 | 21.9 | 20.7 |
| 10 | 238 | | | | 21.4 | 21.8 | 20.3 |

TABLE 1

Results show how, in the case study, the method is very little sensitive to changes in the inter arrival time. The directional sectors are the same for the inter arrival times of 2, 5, 7 and 10 days and differ in 15 degrees for the inter arrival time of 1 day. The 100 year return values are also consistent, with variations of less than 2.1% in all sectors.

CHANGES TO THE MANUSCRIPT

Results in the case study are little sensitive to changes in the inter arrival time and therefore, the above discussion has not been included in the manuscript to keep it concise.

**8. It is not clear to me how the threshold has been chosen (and it would be nice to have indication about CHANGES TO THE results depending on the threshold)**

In this study, the threshold used to define the extreme data was chosen by applying the method in Solari et al. (2017), i.e. the threshold that maximizes the p-value of the Anderson-Darling test, under the assumption that the omnidirectional data come from a Generalized Pareto Distribution.

As shown in Table 2, directional sectors are sensitive to changes in the threshold u (first column), i.e., different definitions of what is an extreme value result in different directional sectors. However, for thresholds above the chosen one, the sectors are quite stable (in particular S2). Therefore, if there is no strong criterion for an a priori selection of the threshold, a sensitivity analysis of the results is recommended. In this situation, p0 (now defined as suggested in (b) in column 9) could serve as an indicator for the final choice of the threshold.

| $u$ | $N_s$ | $S_1$ | $S_2$ | $S_3$ | $u_{100,S1}$ | $u_{100,S2}$ | $u_{100,S3}$ | $p_0$ |
|---|---|---|---|---|---|---|---|---|
| [m/s] | [-] | [deg] | [deg] | [deg] | [m/s] | [m/s] | [m/s] | [-] |
| 14.0 | 306 | 20-140 | 140-195 | 195-20 | 19.9 | 21.5 | 23.0 | 0.9489 |
| **14.3** | **270** | **125-235** | **235-290** | **290-125** | **21.8** | **22.1** | **20.5** | **0.8822** |
| 14.5 | 251 | 145-235 | 235-285 | 285-145 | 22.5 | 22.0 | 20.5 | 0.8799 |
| 15.0 | 203 | 200-235 | 235-290 | 290-200 | 23.0 | 20.7 | 21.5 | 0.8668 |

TABLE 2

*Solari, S., Egüen, M., Polo, M. J., & Losada, M. A. (2017). Peaks Over Threshold (POT): A methodology for automatic threshold estimation using goodness of fit p-value. Water Resources Research, 53(4), 2833-2849.*

CHANGES TO THE MANUSCRIPT

The above considerations have been included in the last paragraph of section 4.2 (lines 3-6, page 19).

***9. in last line of page 7 there is a typo "rfrg"***

We have corrected this typo.

CHANGES TO THE MANUSCRIPT

The wrong word has been deleted (line 19, page 7).

**OTHER CHANGES AND CORRECTIONS**

We have corrected the wrong details in the reference Solari et al. 2017 (lines 30-31, page 23):

[revised manuscript text omitted]